# Epidemiological and Histopathological Characterization of Endometrial Carcinoma: A Retrospective Cohort from Romania

**DOI:** 10.3390/diagnostics15202645

**Published:** 2025-10-20

**Authors:** Andrei Muraru, Alex-Emilian Stepan, Claudiu Margaritescu, Mirela Marinela Florescu, Anne-Marie Badiu, Iulia Oana Cretu, Bianca Catalina Andreiana, Raluca Niculina Ciurea

**Affiliations:** 1Emergency County Hospital of Craiova, 200642 Craiova, Romania; muraru.andrei23@gmail.com (A.M.); raluca.ciurea@umfcv.ro (R.N.C.); 2Anatomic Pathology Department, University of Medicine and Pharmacy of Craiova, 200349 Craiova, Romania

**Keywords:** histopathology, staging, grading, myoinvasion, lymphovascular invasion, perineural invasion, review

## Abstract

**Background/Objectives**: Endometrial carcinoma is an emerging challenge for public health systems globally, especially in countries with a high development index. Traditionally, histopathological staging and grading have been the main criteria informing treatment modalities. More recently, clinically actionable molecular targets have been developed, following observations from the TCGA project and the ProMisE cohort. Although promising, the cost of these methods is an obstacle for some countries that lack well developed theranostics infrastructure in their public systems. This study aimed to contextualize our center’s diagnostic experience from the perspective of histopathological diagnosis. **Methods**: This is a retrospective study that selected 109 cases of already diagnosed endometrial carcinoma from the interval of 2017–2023. We analyzed traditional parameters related to staging and grading, using the FIGO 2009 system as well as basic histological parameters (lymphovascular invasion, perineural invasion, necrosis). Excel and SPSS 26 were used for database management and correlations. Findings were contextualized using the more recent studies that reported on similar parameters. **Results**: Higher-grade tumors were associated with lymphovascular invasion (*p* = 0.04) and lymph node involvement (*p* = 0.0006), as well as deeper myoinvasion (*p* = 0.0018). Myoinvasion (*p* = 0.013) and lymphovascular invasion (*p* = 0.0001) were associated with advanced disease (FIGO III and IV). Our cohort showed a relative paucity (6.5%) of non-endometrioid endometrial carcinoma and presence of lymphovascular invasion (9.2%). Perineural invasion was found in 3 cases with extrauterine involvement. **Conclusions**: Histopathological diagnosis represents an integral component in informing clinical management for endometrial carcinoma and should serve as a means of triage for more expensive molecular techniques. It nevertheless presents reproducibility issues. Further efforts should focus on resolving such issues or possibly introducing less-researched parameters like perineural invasion.

## 1. Introduction

Endometrial cancer is an emerging public health phenomenon with diverse, yet incompletely resolved causal factors. This complexity generates a heterogenous epidemiological profile across the globe [1].

According to GLOBOCAN data from 2022, cancer of the uterine corpus is the 6th most common malignancy in women and the 15th most common cancer overall. The incidence reported in Europe was approximately 125,000 new cases per year with around 30,000 yearly deaths. The overall global trend is towards a greater incidence, especially in countries with a high development index [1].

From the perspective of histology, most uterine corpus malignancies are endometrial carcinomas. The first morphological classification of endometrial carcinomas was proposed in 1983 by Bokhman. This system separated endometrial carcinomas in two types according to histology and observed outcomes. Carcinomas with morphological similarity to native endometrial tissue were classified as type I. Type II carcinomas included carcinomas with higher-grade morphology and subsequently, more aggressive tumor behavior and worse patient outcomes [2].

Subsequent research demonstrated that most type I carcinomas are related to chronic, excessive estrogen exposure, unopposed by progesterone. These carcinomas arise in a predictable sequence from a hyperplasia of the inner uterine lining (endometrium). An early molecular event in the oncogenesis of endometrial carcinoma is a loss of function mutation of the PTEN gene. Additional events guide the progression further towards a malignant phenotype. They usually are low-grade endometrioid carcinomas, albeit with some variety in their presentation [3,4].

In contrast, type II carcinomas are more aggressive, have a lower correlation with chronic estrogen exposure and related conditions. They usually appear in the background of atrophied endometrium, in postmenopausal women. These carcinomas are represented by more aggressive histological types: serous, clear cell, undifferentiated or dedifferentiated carcinomas [3,4,5].

Endometrial carcinoma is staged according to the guidelines of the International Federation of Obstetric Gynecology (FIGO). This risk stratification protocol traditionally used anatomical parameters (regional or distant invasion) coupled with histological criteria (tumor grade according to architecture and cytological atypia) in order to predict tumor behavior and patient prognosis. Cytological grading, according to FIGO, is ascertained by quantifying the proportion of the tumor showing areas of solid, non-squamous growth. Grade 1 tumors have at most a proportion of 5% covered by these areas. The cutoff for grade 2 and grade 3 tumors is 6% and 50%, respectively [4,5].

The current WHO recommendations encourage reporting endometrial carcinomas using a two-tier system where FIGO grades 1 and 2 are considered low-grade endometrial carcinomas. Grade 3 endometrial carcinomas and non-endometrioid, type II carcinomas, are classified as high-grade lesions [4].

More recently, in light of findings from The Cancer Genome Atlas Project and the ProMise cohort, FIGO 2023 introduces a stratification that takes four molecular classes into account, according to mutations that are demonstrated to predict tumor behavior and treatment response. The guidelines present the reporting of molecular classes as strongly recommended if feasible. These observations add complexity to the understanding of endometrial carcinoma, but their predictive power is lacking in some circumstances; however, exploring these is beyond the scope of this work [5,6,7].

Although these new approaches have produced generally reliable categories, classical histology and clinical staging are still indispensable to accurately predict outcome and guide treatment [4,5,6].

With respect to the evolving landscape of the pathology of endometrial carcinoma, our study aims for an epidemiological and morphological characterization of endometrial carcinoma cases obtained at the County Emergency Clinical Hospital of Craiova and also presenting a brief review of the current landscape in the field of histopathological diagnosis.

## 2. Materials and Methods

This work is a retrospective study of 109 cases of endometrial carcinoma diagnosed between 2017 and 2023. The surgical specimens were received from the Obstetrics–Gynecology and General Surgery departments of the County Emergency Clinical Hospital of Craiova. They were received at the time for routine diagnosis and staging according to the parameters of the FIGO (International Federation of Obstetrics and Gynecology) 2009 and the UICC (Union for International Cancer Control) pTNM guidelines, as available at the time of the initial diagnosis. The specimens were fixed in a solution of 10% buffered formalin and processed according to the classical histopathology procedure. The slides were reviewed and relevant microscopic aspects were captured using the Nikon Eclipse E600 microscope, a color CCD camera, and the Lucia 5 software.

Data was collected from the medical records of the Anatomic Pathology Department of the mentioned institution. This allowed for the reporting of patient age, parameters related to staging (anatomical invasion, myometrial invasion, lymph node invasion and metastatic disease), as well as histopathological parameters relating to tumor type and grade, perineural and lymphovascular invasion, and tumor necrosis and squamous metaplasia.

Only resection specimens as part of radical hysterectomies were included, allowing for satisfactory staging of lesions.

Data acquisition and analysis was performed using Excel and the SPSS 26 software package. The chi-square test was used to explore potential correlations. Values of *p* lower than 0.05 were considered significant.

## 3. Results

### 3.1. Epidemiological Data

Our cohort spanned the age range of 35–82 with a mean age at presentation of 62.72 (SD = 9.49). The decades 60–69 and 70–79 contained the most cases, representing 44% and 22.9% of the cohort, respectively. Premenopausal categories, considered as the age intervals of 35–39 and 40–49, represented 13 of the cases (11.9%) (Table 1).

Stratification by tumor grade revealed a majority of grade 1 tumors (*n* = 53; 48.6%). Grade 2 tumors represented 35.7% of the cases (*n* = 39). Grade 3 tumors were identified in 15.7% of the cases (*n* = 17). The first two categories according to stratification by age (30–39; 40–49) yielded predominantly (9/13 cases) low-grade (G1) endometrioid tumors (Table 2; Figure 1).

The oldest two categories (60–69; 70–79) contained the majority of G3 tumors (15/17) total cases). Premenopausal women (30–39; 40–49) developed predominantly low-grade carcinomas, with 11 of the 13 counted cases in these age categories being grade 1 and 2 (Table 2).

Decades 4, 5 and 6 showed a predilection towards lower-grade carcinomas (grade 1 and 2) compared to older cases from decades 7 and 8. By excluding an outlier case of an 82 year old that developed a grade 2 endometrioid carcinoma confined to the uterus, the correlation yielded statistical significance (*p* = 0.047) (Table 2).

We also identified a remarkable case of a serous carcinoma with advanced stage (IVB) at diagnosis in a 35 year old woman.

Regarding the year of diagnosis, there was a discernible downward trend in the interval of 2019–2022. This interval contained only 45 of the total 109 cases (Table 3).

### 3.2. Staging Data

A total of 62.3% of the cases (*n* = 68) were confined to the uterus, corresponding to stage IA and IB according to FIGO 2009 [8]. Stage IA cases represented 28.4% of the cohort (*n* = 31), while 33.9% of cases (*n* = 37) were considered stage IB (Figure 2).

A total of 91 cases were shown to be confined to the uterus (stage IA, IB, II), while 18 were of a higher stage. Non-endometrioid carcinomas were associated with higher stages (*p* = 0.007). The same relationship is true when cases are stratified by tumor grade (*p* = 0.000003) (Table 4, Figure 2).

Cervical stroma involvement was certified as the highest-stage criterion in 21.1% of the cases (*n* = 23) but was present in 24 of the cases in the studied cohort (Figure 1).

Tumor invasion into the serosa or adnexal structures was the highest staging criterion found in eight of the cases. Parametrial involvement was identified in one case (Figure 1).

Pelvic nodes were involved in six cases, while paraaortic nodes were positive in one case. We also found one case invading the sigmoid colon and two cases of metastatic peritoneal disease (Figure 1).

### 3.3. Histopathological Analysis

From the perspective of histology, the following parameters were studied: histological type, myoinvasion (MI), lymphovascular space invasion (LVI), tumor necrosis, perineural invasion (PNI) and squamous metaplasia.

#### 3.3.1. Grade and Histological Type

Regarding histological type, 93.5 of tumors (*n* = 102) were endometrioid tumors of varying grades. Among these, 48.6% tumors (*n* = 53) were graded as FIGO grade 1 (Figure 2). Grade 2 tumors accounted for 35.7% of the cohort (*n* = 39) (Figure 3), while grade 3 endometrioid carcinoma was found in 10 cases (9.1%) (Figure 4 and Table 5).

Well-differentiated tumors (grade 1 and 2) contain glands similar to proliferative phase endometrium with varying degrees of secretory and mucinous differentiation. Our cohort contained a remarkably well differentiated carcinoma with uniform secretory differentiation and a broad pushing front.

Grade 3 endometrioid tumors were more morphologically heterogeneous, presenting mostly as solid areas with subtle gland formation. Two of the cases contained a tubular/papillary morphology reminiscent of serous carcinoma but with no accompanying cytological atypia as would be expected from serous carcinomas.

Non-endometrioid carcinomas were identified in seven cases (6.5%). Of these, three cases (2.8%) were carcinosarcomas and two cases (1.9%) were represented by serous carcinoma, with one case each (0.9%) of clear cell carcinoma and mixed carcinoma, respectively (Table 5, Figure 5).

#### 3.3.2. Histopathological Parameters

Four of the cases (3.7%) were confined to the endometrium. Myoinvasion (MI) was detected in the remainder of 105 cases, with 41 cases (37.6%) showing invasion of the first half. Invasion involving the outer half of the myometrium was detected in 58.7% of cases. Depth of invasion was directly correlated to tumor grade with high-grade tumors showing a greater depth of invasion. The relationship was statistically significant (*p* = 0.02). Depth of MI was correlated with higher FIGO stages (*p* = 0.013) (Table 6, Figure 6).

Lymphovascular invasion (LVI) was detected in 10 (9.2%) of cases. All of the tumors with confirmed LVI also presented with invasion of the second half of the endometrium. This parameter was positively correlated with higher tumor grades (*p* = 0.02). LVI was also correlated with MI (*p* = 0.01), as were higher FIGO stages (*p* = 0.02) (Table 6, Figure 7).

Considerable areas of tumor necrosis were identified in 15 cases (13.7%) with no significant correlations found with most histological parameters, age or FIGO stage. It was, however, correlated to LVI (*p* = 0.01) (Table 6, Figure 7).

Perineural invasion (PNI) was identified in only three cases (2.8%), while tumor necrosis was identified in 15 cases (13.8%). Interestingly, all of the cases with positive PNI presented at stage IIIA (serosal or adnexal involvement) (Table 6, Figure 7).

Squamous and morular metaplasia was identified in 39 of the cases (35.7%). It was found mostly in lower-grade endometrioid carcinomas (grade 1 and 2). No significant relationships with other histological parameters were identified (Table 6, Figure 7).

Lymph node invasion (LNI) could not be accurately assessed in 67 of the cases (61.4%). Of the remainder, lymph nodes were negative in 35 cases (32.1%) and positive in 7 cases (6.5%). Lymph node status was directly correlated to grade, with higher tumor grades showing more LNI (*p* = 0.00002). LNI was also associated with the presence of LVI (*p* = 0.01) (Table 6).

The most significant relationships between histopathological parameters are highlighted below (Table 7). The findings are also summarized in relation to contemporary data in the Appendix (Table A1).

## 4. Discussion

The growing incidence rate of endometrial carcinoma is of particular interest in recent years. It is the 4th most common cancer affecting women in high socioeconomic index nations. This highlights the necessity of adopting prevention and screening programmes [1,5].

Endometrial carcinoma represents a complex taxonomy of biologically diverse tumors with a wide morphological spectrum and diverse clinical behavior. The long-term stimulating effect of estrogen on endometrial cells is considered the most important inciting factor [3,4].

The mean age reported in recent cohorts varies between 54 and 64.9 [9,10]. The mean age of our cohort places it in the upper range of contemporary epidemiological studies.

According to Hoang et al., in a study evaluating the reproducibility of reporting tumor grade, grade 1 tumors show the least interobserver variability [11]. Our cohort finds grade 1 tumors in 48.6% of the cases. Recent studies show a range between 34.3% and 59.2% for this finding [12,13].

Grade 2 tumors represented 35.7% of our cohort, with other recent cohorts showing a range of percentages between 23.1 and 51.4% [12,13].

The more recent WHO guidelines encourage reporting endometrial carcinoma using a two-tiered classification system, in which grade 1 and grade 2 tumors are considered well-differentiated and grade 3 tumors are considered poorly differentiated [4]. The segregation is justified by the fact that grade 3 tumors have a worse overall prognosis and it is replicated in recent prospective studies [14,15,16].

This difference in outcomes is reflected in the current study, upon taking into consideration histopathological parameters that are already established as predictors for poor overall survival or shorter disease-free progression. Our study finds a direct correlation between lymphovascular space invasion and grade 3 tumors (*p* = 0.04) and higher FIGO stages (*p* = 0.0001), respectively.

Regarding FIGO staging, data from our cohort is represented mostly by tumors confined to the uterus, with 83.5% of the cases being staged as FIGO I and II. Data in the literature show an interval between 45.7% and 87% of carcinoma cases confined to the uterus [15,17]. Our study does not discern between the effects of different screening practices or other population-level dynamics.

A particular difficulty in reporting endometrial carcinoma is discerning between grade 3 endometrioid tumors and non-endometrioid entities [4].

A study from 2017 by Thomas et al. included 131 cases of grade 3 carcinomas for review by two expert Gynecologic Pathologists from two different institutions. The authors reclassified 33 of the cases into other diagnostic categories as part of the review. The reported diagnostic agreement between each of the two institutions and the expert reviewers was 83% and 62%, respectively. The most important categories obtained through reclassification were undifferentiated carcinoma (7/131) and serous carcinoma (5/131) [18].

Discerning between high-grade endometrioid carcinomas and non-endometrioid carcinomas is particularly important, as these two entities have a remarkably different natural progression and require different treatment approaches [19].

Non-endometrioid carcinomas represent 6.5% of our cohort, of which two were serous carcinomas and one a case of mixed-cell carcinoma (cumulative 2.8%).

Historically, and also supported by recent data from SEER, serous carcinoma has had a higher incidence in African Americans. Women from this group have up to a fourfold increase in incidence compared to other groups in the US, as shown by Abel et al. [20] It is still unclear if this is a feature derived from genetic or socio-economic factors.

Montoya et al. find the maximum proportion of serous carcinomas as part of a cohort gathered from a Colombian population. They report a cumulated proportion of 2.9% of cases as serous or mixed-cell carcinoma. The same study nonetheless reports 17.5% of the cases as unclassifiable [17].

The data from a case conducted in Nigeria by Olatunde et al. finds a proportion of 20.5% of cases as non-endometrioid, with serous carcinoma reported as 18.2% of their cohort. It is important to take into account the reproducibility difficulties encountered in this category, as discussed above [21].

A recent US cancer registry study by Saris et al., using the SEER database, finds a higher proportion of serous carcinoma (18.4%) and carcinosarcoma (14.1%) in the African American subgroup of the cohort. By contrast, the non-African American subgroup showed an incidence for the same entities of 7 and 4.7%, respectively [22]. A recent study by Guttery et al., using tumor material gathered from a US cohort, shows a higher relative rate of somatic mutations of TP53, in the African American subgroup, regardless of histological type. The same study finds a rate of PTEN somatic mutations of 63% and 85% for the Caucasian and Asian subgroups, respectively. PTEN mutations are associated with endometrioid carcinoma rather than serous variants [23].

Furthermore, evidence from single-nucleus sequencing of serous carcinoma tissue specimens show up-regulation of pathways involved in aggressiveness and local immunosuppression. This relationship was found only in the African American subgroup of this particular study by Foley et al. [24].

Burney et al. found the highest incidence of non-endometrioid carcinoma (28%) in a recent cohort from Oman. The group does not, however, report on the incidence of serous carcinoma or the ethnic composition of their cohort [25].

The relationship between ethnicity and serous histotypes has so far not reached a consensus. Our study cohort does not contain ethnic composition data. The relative paucity of non-endometrioid carcinomas in our cohort may in part be explained by this dynamic.

Lymphovascular invasion is an important histological predictor. It has been found in association with lymph node metastasis. It is shown as an independent risk factor to lymph node recurrence and distant metastasis and is important in formulating adjuvant treatment [26].

Our study reported LVI as present in 9.2% of cases. We found associations between lymphovascular invasion and the depth of myometrial invasion (*p* = 0.01), higher FIGO grades (*p* = 0.02) and higher FIGO stages (*p* = 0.02).

A recent study by Li et al. finds that, in regard to prognostic data, focal LVI has similar outcomes to no LVI [27]. Substantial LVI (>5 vessels) is shown to strongly impact patient overall survival and progression-free survival and is significant for staging in the FIGO 2023 guidelines [6]. Lymphovascular invasion is more commonly seen in type II carcinomas [28]. Its predictive power for survival is weaker for low-grade endometrioid carcinomas with node-negative status, as shown by Pifer et al. [29].

Recent data obtained from studies with a similar design find that up to 36.5% of cases contain LVI, although not qualifying the extent. The mentioned study contained 21.5% non-endometrioid tumors as part of their cohort and 33.5% of cases with advanced FIGO stages, possibly explaining the relatively high proportion of cases with LVI in their cohort [16].

Conversely, our cohort contained a majority of low-grade endometrioid carcinomas (74.3%) which would explain the relatively low proportion of LVI reported.

Although it is strongly associated with reduced overall survival as an independent parameter in recent studies with prospective wings, LVI has not been standardized between the more recent reporting guidelines [30,31].

Myometrial invasion was found in 96.3% of our cohort with 58.7% of cases showing invasion past the first half of the myometrium. Data in the literature for MI past the first half of the myometrium ranges from 28.2% to 53.5% [9,15].

Higher depth of invasion correlates with higher FIGO stages and higher architectural and cytological grade and is replicated in our cohort [6].

Positive lymph nodes are determinants of higher stages in endometrial carcinoma. FIGO 2009 staging contains the category of IIIC1, corresponding to positive pelvic nodes. The IIIC2 category corresponds to positive para-aortic lymph nodes, while stage IVB is reported for positive inguinal nodes [6].

Our study cohort finds positive lymph nodes in 6.5% of cases, with 4 cases reported as IIIC1, 2 as IIIC2 and 1 as IVB case, respectively.

Data in the reviewed studies ranges from 6.4% to 13.5% for cases with at least 1 unspecified lymph node involved [14,15].

The majority of the studies did not report on the Nx category. As a limitation, our study also included incidentally diagnosed endometrial carcinoma with subsequent incomplete lymph node exenteration, affecting the Nx category.

A population-based study from Germany by Papathemelis et al. reports this finding in 9.4% of cases, with only 12% of cases identified in the Nx category [32].

Perineural invasion is a mode of invasion present in most cancers. It was found in our cohort in three FIGO IIIA cases. To our knowledge, there are no epidemiological data referring to perineural invasion in endometrial cancer. Although it is demonstrated in a variety of abdomino-pelvic carcinomas, its role in endometrial carcinoma has not been researched [33]. It is however demonstrated histologically in the context of endometriosis-associated pain [34].

Furthermore, an in vitro study using endometrial carcinoma cell lines illustrates the ability of these cells to engage in perineural invasion through paracrine crosstalk involving nerve-associated cells [35].

To date, there are no recommendations for reporting perineural invasion in endometrial carcinoma.

## 5. Conclusions

The landscape of endometrial carcinoma is a rapidly evolving field that brings together findings from multiple domains of research. In line with the complexity of the endometrium in physiological conditions, endometrial carcinoma is conceptualized as an umbrella category of diverse tumors with equally diverse behavior and prognosis.

Even if more recent research efforts have been focused on categorizing the lesions by using data from molecular studies, histology remains a robust indicator of outcome and is indispensable as a first step in translational approaches. This is especially true for centers with lower access to molecular infrastructure.

Perineural invasion lacks support in the literature regarding incidence in epidemiological studies, with none published to date. It nevertheless might represent a future area of interest, in light of more recent in vitro studies.

Furthermore, a precise understanding of the morphological spectrum of these entities is required so that practicing pathologists can more accurately inform the necessity of additional testing in an economical manner. Discerning between presentations of high-grade carcinomas through histopathology remains a diagnostic limitation. The large variability of reported incidence of non-endometrioid carcinomas, which is also apparent in our study, might be a factor of observer error, given the lack of supporting ancillary tests.

## Data Availability

The original contributions presented in this study are included in the article. Further inquiries can be directed to the corresponding author.

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
