# Peer review of "Epidemiological and Histopathological Characterization of Endometrial Carcinoma: A Retrospective Cohort from Romania"

_diagnostics, 2025, doi:10.3390/diagnostics15202645_

Round 1

Reviewer 1 Report

Comments and Suggestions for Authors

The authors discuss one of the most interesting and challenging neoplasms of the female genital tract. The detailed analyses and the accompanying literature review substantially increased the positive impact of this study, providing to the readers resources to study endometrial lesions with refined and informative scientific support. The microphotographs are of excellent quality, and the Discussion section reveals the authors' considerable scientific maturity. It is important, however, to add details about the manufacturer and model of the Motic Panthera E2 system, as not all readers are familiar with the device and its usefulness in laboratory practice.

Author Response

We thank you for your comments and the attention paid for our submission!

We updated the information in the material and methods section according to your suggestion.

It was, sadly, a slip in editing from an earlier draft where we used a newer, albeit less powerful device. The manuscript is updated with the correct platform "Nikon e600 using a CCD camera and the Lucia 5 software". 

Reviewer 2 Report

Comments and Suggestions for Authors

I have had the opportunity to review the manuscript "Difficulties in diagnosing endometrial carcinoma- experience from a teaching hospital in southeastern Romania." The study is timely and reflects the real-world challenges in the diagnosis of endometrial cancer, making it valuable for professionals working in similar settings.
The manuscript addresses an important topic, and the experience presented is highly relevant to many clinics and teaching hospitals. he authors conducted a detailed analysis of a small cohort of cases, which yielded valuable data. The micrographs used in the study are of good quality, effectively illustrating the morphological features of the tumors. 
Despite its strengths, the manuscript requires significant revisions to meet current scientific standards and be a complete scientific publication.

  1. The current title "Difficulties in diagnosing endometrial carcinoma- experience from a teaching hospital in southeastern Romania" is too narrow and may limit the audience. The title should be more informative and appealing, reflecting the clinico-pathological aspects of the study. I recommend that the authors consider options that include key diagnostic markers or novel approaches used in the work.
  2. Although the introduction contains a historical overview, it lacks sufficient information about the current state of classification and diagnosis of endometrial cancer. The authors should expand this section to show how their study fits into the context of current scientific discussions.
  3. Use of Outdated Classification: This is the most critical comment. The use of the 2009 classification is unacceptable. Since its publication, new, molecularly-oriented approaches to classifying endometrial cancer have been developed. The authors must reclassify all cases according to the latest international recommendations (not at the moment of the routine diagnosing). This will significantly increase the scientific value of the work and allow its results to be compared with other modern studies.
  4. The "Materials and Methods" section needs to be expanded. A more detailed description of the morphological (histological) studies used is necessary. This will allow other researchers to reproduce the experiment and better understand the context of the results obtained.

Author Response

We thank you for the effort in reviewing our submission!

  1. Comment 1:The current title "Difficulties in diagnosing endometrial carcinoma- experience from a teaching hospital in southeastern Romania" is too narrow and may limit the audience. The title should be more informative and appealing, reflecting the clinico-pathological aspects of the study. I recommend that the authors consider options that include key diagnostic markers or novel approaches used in the work.

We have updated the title that more accurately describes the scope of the manuscript, while also mentioning geographic data.

      2. Comment 2 and 3:

  1. Although the introduction contains a historical overview, it lacks sufficient information about the current state of classification and diagnosis of endometrial cancer. The authors should expand this section to show how their study fits into the context of current scientific discussions.
  2. Use of Outdated Classification: This is the most critical comment. The use of the 2009 classification is unacceptable. Since its publication, new, molecularly-oriented approaches to classifying endometrial cancer have been developed. The authors must reclassify all cases according to the latest international recommendations (not at the moment of the routine diagnosing). This will significantly increase the scientific value of the work and allow its results to be compared with other modern studies.

We are aware of the revolution regarding molecular classification in this area. As of now, the oncologists we work with, have not pressed us to deliver investigations of the kind (NGS or POLEmut sequencing namely) and as such the infrastructure in our center is inexistent. To our knowledge there are only 2 private laboratories in romania that can work POLEmut on paraffin embedded tissue. 

At the same time, the scientific ecosystem right now voices critique around the changes. Namely around cost, reproducibility and actionability of the findings. Even the original publication of Berek for 2023 mentions that molecular classification should be employed if feasible.

With that said, we are not arguing that employment of such methods is not important. Rather the fact that before such changes are implemented, with all their associated costs, pathologists worldwide should focus on the reproducibility gap around entities like high grade carcinomas or LVI.

FIGO 2023 endometrial cancer staging: too much, too soon? - International Journal of Gynecological Cancer

New FIGO 2023 Staging System of Endometrial Cancer: An Updated Review on a Current Hot Topic - PMC

FIGO staging of endometrial cancer: 2023 - Berek - 2023 - International Journal of Gynecology & Obstetrics - Wiley Online Library

Comment 4: 

  1. The "Materials and Methods" section needs to be expanded. A more detailed description of the morphological (histological) studies used is necessary. This will allow other researchers to reproduce the experiment and better understand the context of the results obtained.

The cases were drawn from the archive of cases diagnosed in that period. They were reviewed to ensure adherence to the histological categories and staging criteria appropriate for the context. Could you please clarify on what we can expand? Thank you very much for your effort.

Reviewer 3 Report

Comments and Suggestions for Authors

Review Report

Title: “Difficulties in diagnosing endometrial carcinoma- experience from a teaching hospital in southeastern Romania

Authors: Andrei Muraru, Alex-Emilian Stepan, Margaritescu Claudiu, Mirela Florescu, Badiu Anne-Marie, Cretu Iulia, Bianca Andreiana, Raluca Niculina Ciurea.

Comments:

In their paper “Difficulties in diagnosing endometrial carcinoma- experience from a teaching hospital in southeastern Romania” Muraru and colleagues aimed at providing an epidemiological and morphological characterization of endometrial carcinoma cases obtained at the County Emergency Clinical Hospital of Craiova and presenting a brief review of the current landscape in the field of histopathological diagnosis.

The article is a well-designed retrospective study, showing a background about histopathological diagnosis of endometrial carcinoma and clarifying how this relates to their center’s diagnostic experience, which can help in better understanding. They selected 109 cases of already diagnosed endometrial carcinoma. They analyzed traditional parameters related to staging and grading, using the FIGO 2009 system as well as basic histological parameters (lymphovascular invasion, perineural invasion, necrosis). They used Excel and SPSS 26 for database management and correlations. They contextualize their findings by linking them to the more recent studies that reported on similar parameters. They found that higher grade tumors were significantly associated with lymphovascular invasion and lymph node involvement, as well as deeper myoinvasion. In addition, they demonstrated that myoinvasion and lymphovascular invasion were significantly associated with advanced disease (FIGO III and IV).

Taken together, the study is well conducted and adds substantial new knowledge regarding

Histopathological diagnosis represents an integral component in informing clinical management for endometrial carcinoma and should serve as a means of triage for the more expensive molecular techniques. It nevertheless presents reproducibility issues. They highlighted the need for further efforts for resolving such issues or studying less researched parameters like perineural invasion. Consequently, these findings could have considerable clinical impact by improving diagnosis and treatment of endometrial carcinoma.

The authors should, however, do some minor additional modifications to the manuscript:

  • Results section: It is better to re-write this section in a more organized way and divide it into subsections and put heading for each subsection. This will be better for the reader to follow the results.
  • The authors created multiple tables for their univariate analysis, they can create one table including all categories like Age, Grade, stage. And also including P values.
  • The authors keep mentioning Panel 1 to Panel 7 and they did not put any title for these panels under the H&E images. They have to write under each Panel a title including the number of the panel and what this panel shows.

Author Response

We thank you for your time and effort in reviewing our submission.

  • Results section: It is better to re-write this section in a more organized way and divide it into subsections and put heading for each subsection. This will be better for the reader to follow the results.
  • The authors created multiple tables for their univariate analysis, they can create one table including all categories like Age, Grade, stage. And also including P values.
  • The authors keep mentioning Panel 1 to Panel 7 and they did not put any title for these panels under the H&E images. They have to write under each Panel a title including the number of the panel and what this panel shows.

As to your suggestions, we have worked on point 1 and 3 of your review report. Sadly to those mentioned in point 2 of your comment, 3 of the other reviewers complained on the readability on the manuscript. We have a table in the appendix that summarizes the findings and contextualizes them with recent data. If you feel that is an important point to hit, we will gladly approach it in round 2 of reviews. Thank you very much again

Reviewer 4 Report

Comments and Suggestions for Authors

Abstract:

  • overall well presented
  • the word "prognosticators" although it could stay in the manuscript, i would suggest the authors to change it to a term more commonly used in scientific papers, like prognostic factor, indicator of outcome.... 

Introduction: 

  • well presented, i have no further suggestions 

Materials and methods: 

  • well presented, i have no further suggestions 

Results: 

  • "Regarding the year of diagnosis, there was a discernible downward trend of the frequency of diagnosis for the interval of 2019-2022". - this is an interesting information, it would be good to comment on why you think this happened since the incidence of endometrial Ca is on the rise
  • Figure 1 and Table 2 - some numbers moved around in editing, it should be fixed 

Discussion: 

  • They instead find a positive correlation with more advanced stages at diagnosis. [19, line 363 - correlation with lower of higher socioecenomic status?
  • there are a lot of references about different races and ethnicities, are they included in you cohort? if so, you should state that, otherwise those paragraphs, even though it is interesting information, should be summed up to be shorter

Conclusion: 

  • well presented, i have no further suggestions 

Author Response

We thank you for the attention lent to our manuscript submission. The editing error was fixed, the layout is clear now hopefully throughout. We modified prognosticator to different terms around the manuscript.

As to the first comment:

"Regarding the year of diagnosis, there was a discernible downward trend of the frequency of diagnosis for the interval of 2019-2022". - this is an interesting information, it would be good to comment on why you think this happened since the incidence of endometrial Ca is on the rise"

Three of the other 5 reviewers voiced concern over the readability and volume of the manuscript at points. The structure of the manuscript will not allow making that comment inside the results section and the discussion section is already dense. If you feel this is an important point to hit, we will gladly include this in round 2 revisions. It's most likely an effect of COVID but its too speculative to infer with certainty from the standpoint of a retrospective study with no access to how surgeons actually scheduled their interventions.

As to the other 2 major comments:

  • They instead find a positive correlation with more advanced stages at diagnosis. [19, line 363 - correlation with lower of higher socioecenomic status?
  • there are a lot of references about different races and ethnicities, are they included in you cohort? if so, you should state that, otherwise those paragraphs, even though it is interesting information, should be summed up to be shorter

We removed the line referencing the populational study from Columbia as it hindered the argument we were trying to construct.

Namely, and to your second comment, mentions of the racial and economic classes, were made in an effort to contextualize the reproducibility of the serous carcinoma category worldwide. We have revised said paragraphs to your and another editor's suggestions. We hope they are clearer right now! 

Thank you again and feel free to voice any other concerns.

Reviewer 5 Report

Comments and Suggestions for Authors

Review of the manuscript Diagnostics 30.09.2025

The article “Difficulties in diagnosing endometrial carcinoma – experience from a teaching hospital in southern Romania” is logically structured and well-organized. Research on endometrial cancer is inherently valuable. However, in its current form, the manuscript is not suitable for publication. It requires substantial revision, particularly in terms of English language, clarity, and adherence to journal guidelines. There are multiple critical issues that must be addressed. While the manuscript contains valuable data and is logically organized, it requires major revisions, most importantly a thorough improvement of the English language. Without substantial language editing, the manuscript is not suitable for publication, as poor readability and unclear phrasing compromise the scientific content.

Major concerns

  1. English language and writing quality: The manuscript requires major English language editing. There are numerous stylistic and grammatical errors, unclear sentences, and frequent repetitions (e.g., L15, L43, L90). Without thorough language revision, the manuscript remains difficult to read and understand. Improving clarity, sentence structure, and scientific phrasing is essential before any further review.
  2. Readability and appearance of figures and tables: Figures and tables lack clarity, are inconsistent, and do not conform to the journal’s guidelines. These should be revised and standardized.
  3. Histopathological image panels: Panels are presented inconsistently, and captions take up excessive space, affecting readability. Standardization throughout the manuscript is required.
  4. Clarity of sentences and content: Several sentences are unclear or redundant and need rephrasing. Examples include L75–76, L85–86 (molecular classification is no longer optional in EC management), L135–136 (repetition), and L165–167. Many other sentences throughout the manuscript require careful editing for clarity and readability.
  5. Statistical notation: p-values should be italicized, with spaces around the equality sign, and consistently formatted throughout the manuscript.
  6. Percentage reporting: Percentages should be consistently reported, either to one or two decimal places. Discrepancies are noted (e.g., L281 and L285).
  7. References: Citations should be in square brackets using commas, with periods placed after the bracket. In-text citation style should follow the Author et al. convention.
  8. Terminology: LVI and PNI should be defined as lymphovascular invasion and perineural invasion, not “involvement.”
  9. Interpretation of incidence data: Lines 306–309 should not claim a direct causal link between EC incidence and socioeconomic status; the argument is overly simplified and unsupported by the data.
  10. Statistical details: Percentages should include absolute numbers and, where relevant, p-values for significance (e.g., L333).

Minor concerns

  1. Patient age groups should specify “years” to avoid ambiguity.
  2. L383–384: Clarify whether “single-nucleus sequencing” is meant. L357: The phrase “literature presents a maximum proportion” is unclear.
  3. L417: Typographical error in “myometrium.” L452: missing “is.”
  4. L427–430: Sentence too complex; consider splitting. L442–443: Could be rephrased for clarity.
  5. The list of abbreviations does not include all terms used in the manuscript.

Comments on the Quality of English Language

English language and writing quality: The manuscript requires major English language editing. There are numerous stylistic and grammatical errors, unclear sentences, and frequent repetitions (e.g., L15, L43, L90). Without thorough language revision, the manuscript remains difficult to read and understand. Improving clarity, sentence structure, and scientific phrasing is essential before any further review.

Author Response

We thank you for your commitment on ensuring the quality of the manuscript we submitted. 

Towards the quality of the writing and the readability of the manuscript, we have taken your suggestions seriously and updated the text. We have appropriately formatted according to your suggestions. We will also opt for layout editing offered by MDPI after the second round of reviews.

As to point 4: Clarity of sentences and content: Several sentences are unclear or redundant and need rephrasing. Examples include L75–76, L85–86 (molecular classification is no longer optional in EC management)

We are aware of the developments in this area. We are sadly missing the infrastructure. To our knowledge there is only one private laboratory setup to work POLE mutation status on paraffin embedded blocks. Oncologists as of yet have not requested results of this sort, even if we have theranostics capabilities in more established areas.

With that said, we are also aware of critique of said implementation, even from centers with, expectedly, better material support. The original publication by Berek called the implementation of molecular techniques as encouraged if feasible.

New FIGO 2023 Staging System of Endometrial Cancer: An Updated Review on a Current Hot Topic - PMC

FIGO 2023 endometrial cancer staging: too much, too soon? - PubMed

FIGO staging of endometrial cancer: 2023 - Berek - 2023 - International Journal of Gynecology & Obstetrics - Wiley Online Library

With that said, we are not arguing around the utility of molecular methods. What we are arguing for is that before such efforts are implemented, the pathologist community worldwide needs to seriously address the reproducibility issue of histopathological categories. We would have appreciated a dataset that contained molecular data but we think as of now this work stands on its own, even if it's a minor contribution.

As to point 9 of your report:

  1. Interpretation of incidence data: Lines 306–309 should not claim a direct causal link between EC incidence and socioeconomic status; the argument is overly simplified and unsupported by the data.

We do not infer incidence or correlation in that phrase. We simply used GLOBOCAN's World Bank Classification of countries (filter) to segregate the data, which we think is appropriate, accounting for the prospective audience of the manuscript.

Once again we appreciate your careful efforts. We have touched on every point in your review report and the more important modifications around phrasing have been highlighted in text. We wish you all the best.

Round 2

Reviewer 3 Report

Comments and Suggestions for Authors

Review Report

Title: “Epidemiological and histopathological characterization of endometrial carcinoma: a retrospective cohort from Romania

Authors: Andrei Muraru, Alex-Emilian Stepan, Margaritescu Claudiu, Mirela Florescu, Badiu Anne-Marie, Cretu Iulia, Bianca Andreiana, Raluca Niculina Ciurea.

Comments:

In their paper “Epidemiological and histopathological characterization of endometrial carcinoma: a retrospective cohort from Romania” Muraru and colleagues aimed at providing an epidemiological and morphological characterization of endometrial carcinoma cases obtained at the County Emergency Clinical Hospital of Craiova and presenting a brief review of the current landscape in the field of histopathological diagnosis.

The article is a well-designed retrospective study, showing a background about histopathological diagnosis of endometrial carcinoma and clarifying how this relates to their center’s diagnostic experience, which can help in better understanding.

Taken together, the study is well conducted and adds substantial new knowledge regarding Histopathological diagnosis represents an integral component in informing clinical management for endometrial carcinoma and should serve as a means of triage for the more expensive molecular techniques. It nevertheless presents reproducibility issues. They highlighted the need for further efforts for resolving such issues or studying less researched parameters like perineural invasion. Consequently, these findings could have considerable clinical impact by improving diagnosis and treatment of endometrial carcinoma.

The authors responded well for my previous review report and carried out the suggested modifications.

The authors should, however, do some minor additional modifications to the manuscript:

  • In the introduction, there are some short paragraphs related to the same point, it is better to connect these paragraphs in one paragraph. For Example, the authors added a new paragraph in the line 73 starting with Cytological grading, according to FIGO. It is better to add this new sentence to the previous paragraph staring in line 68, not to create a new paragraph as the two paragraphs related to the FIGO criteria.

Author Response

We thank you for your attention. We have addressed your request. 

Reviewer 5 Report

Comments and Suggestions for Authors

As a reviewer, I must state that, in its current form, I cannot recommend this manuscript for publication. The paper requires substantial revision, as the authors have not addressed the comments and suggestions provided in the previous review. I strongly encourage the authors to carefully reconsider all the feedback and make the necessary improvements to meet the scientific and editorial standards expected by the journal.

Comments on the Quality of English Language

The English language of the manuscript requires significant improvement. Numerous grammatical and stylistic errors hinder the clarity and readability of the text. I recommend that the authors have the manuscript thoroughly revised by a proficient English speaker or a professional language editing service before resubmission.

Author Response

We greet you and thank you for the care in encouraging the quality of our submission.

As to point 1:

  1. English language and writing quality: The manuscript requires major English language editing. There are numerous stylistic and grammatical errors, unclear sentences, and frequent repetitions (e.g., L15, L43, L90). Without thorough language revision, the manuscript remains difficult to read and understand. Improving clarity, sentence structure, and scientific phrasing is essential before any further review.

We have edited most of the discussion chapter with respect also to your suggestion laid out in point 7 regarding       the “Author et al” convention.

As to point 2 and 3:

  1. Readability and appearance of figures and tables: Figures and tables lack clarity, are inconsistent, and do not conform to the journal’s guidelines. These should be revised and standardized.
  2. Histopathological image panels: Panels are presented inconsistently, and captions take up excessive space, affecting readability. Standardization throughout the manuscript is required.

        We are not aware of how the figures and tables veer away from the guidelines. If you consider they are not                correctly captioned or explained please let us know. They now respect the body of the text throughout the         manuscript and we are to expect additional copy editing will be done if the manuscript is accepted. In case the    concern refers to an uneven distribution of elements in the panel, let us know.

As to point 4:

  1. Clarity of sentences and content: Several sentences are unclear or redundant and need rephrasing. Examples include L75–76, L85–86 (molecular classification is no longer optional in EC management), L135–136 (repetition), and L165–167. Many other sentences throughout the manuscript require careful editing for clarity and readability.

        We feel this has been adequately addressed. If not, let us know.

  1. Statistical notationp-values should be italicized, with spaces around the equality sign, and consistently formatted throughout the manuscript. - The formatting changes were made across the manuscript.
  2. Percentage reporting: Percentages should be consistently reported, either to one or two decimal places. Discrepancies are noted (e.g., L281 and L285). - The formatting changes were made across the manuscript by using a single decimal throughout.
  3. Terminology: LVI and PNI should be defined as lymphovascular invasion and perineural invasion, not “involvement.” - This has been standardized through using “invasion”.
  4. Interpretation of incidence data: Lines 306–309 should not claim a direct causal link between EC incidence and socioeconomic status; the argument is overly simplified and unsupported by the data. - This has been addressed in an earlier response. If you suggest introducing a phrase that reflects we used the world bank country dataset inside GLOBOCAN, let us know.
  5. Statistical details: Percentages should include absolute numbers and, where relevant, p-values for significance (e.g., L333). - They were consistently used to this standard inside the results section, if you express the preferrence to keep the standard for the discussion section, let us know and we will comply.

Minor concerns

  1. Patient age groups should specify “years” to avoid ambiguity.
  2. L383–384: Clarify whether “single-nucleus sequencing” is meant. L357: The phrase “literature presents a maximum proportion” is unclear.
  3. L417: Typographical error in “myometrium.” L452: missing “is.”
  4. L427–430: Sentence too complex; consider splitting. L442–443: Could be rephrased for clarity.
  5. The list of abbreviations does not include all terms used in the manuscript.

        All minor concerns have been addressed during the first round. We take this time to apologize for not considering how the quality of our response can hinder the tracking of the changes made. We wish you all the best and we are more than willing to make future modifications if need be.